# Forgotten but not gone: A multi-state analysis of modern-day debt imprisonment

**Johann D. Gaebler**[1]*, **Phoebe Barghouty**[2], **Sarah Vicol**[3], **Cheryl Phillips**[2], **Sharad Goel**[4]

1 Department of Statistics, Harvard University, Cambridge, MA, United States of America, 2 Communication Department, Stanford University, Stanford, CA, United States of America, 3 Department of Economics, Stanford University, Stanford, CA, United States of America, 4 Harvard Kennedy School, Harvard University, Cambridge, MA, United States of America

* jgaebler@fas.harvard.edu

## Abstract

In almost every state, courts can jail those who fail to pay fines, fees, and other court debts— even those resulting from traffic or other non-criminal violations. While debtors' prisons for private debts have been widely illegal in the United States for more than 150 years, the effect of courts aggressively pursuing unpaid fines and fees is that many Americans are nevertheless jailed for unpaid debts. However, heterogeneous, incomplete, and siloed records have made it difficult to understand the scope of debt imprisonment practices. We culled data from millions of records collected through hundreds of public records requests to county jails to produce a first-of-its-kind dataset documenting imprisonment for court debts in three U.S. states. Using these data, we present novel order-of-magnitude estimates of the prevalence of debt imprisonment, finding that between 2005 and 2018, around 38,000 residents of Texas and around 8,000 residents of Wisconsin were jailed each year for failure to pay (FTP), with the median individual spending one day in jail in both Texas and Wisconsin. Drawing on additional data on FTP warrants from Oklahoma, we also find that unpaid fines and fees leading to debt imprisonment most commonly come from traffic offenses, for which a typical Oklahoma court debtor owes around $250, or $500 if a warrant was issued for their arrest.

## Introduction

Courts in the United States impose significant financial burdens on defendants, through fines, fees, restitution, and other court debts (e.g., [1]). Brick-and-mortar debtors' prisons disappeared from the American legal system in the mid-19th century as their perception as inhumane and ineffective institutions grew [2]. But the threat of jail only died out for unpaid private debts. In the overwhelming majority of states, failure to pay court debts is punishable by imprisonment—even for debts arising from traffic or other minor violations not themselves punishable by imprisonment [3]. In theory, court debtors enjoy a range of constitutional and statutory protections from incarceration unless they "willfully" fail to pay [4]. However, a number of high-profile investigations have uncovered jurisdictions like Ferguson, Missouri [5]; Corinth, Mississippi [6]; and Jackson, Mississippi [7], where courts have regularly imprisoned court debtors, often with little regard for ability to pay.

**Data Availability Statement:** The data underlying the results presented in the study are available from policylab.stanford.edu/debtors-prisons.

**Funding:** This study was supported by a grant from Arnold Ventures (https://www.arnoldventures.org). The funders had no role in study design, data collection and analysis, decision to publish, or preparation of the manuscript.

**Competing interests:** The authors have declared that no competing interests exist.

These reports point to a reemergence of widespread incarceration for failure to pay court—rather than private—debts, and raise questions about debt imprisonment's true prevalence and impacts. However, a combination of factors has stood in the way of a systematic understanding of incarceration for failure to pay. As a result of the decentralized nature of American law enforcement and legal and policy differences across states and counties, there are not comprehensive and standardized records documenting incarceration practices, including the practice of debt imprisonment. In particular, there is no national dataset documenting debt imprisonment—or even statewide studies, except in Rhode Island [8].

Drawing on a novel dataset we collected from hundreds of public records requests filed with local law enforcement officials across the United States, this study aims to provide credible order-of-magnitude estimates of the prevalence of debt imprisonment and some of its important consequences. In particular, we estimate the *per capita* rate at which residents of Texas and Wisconsin are jailed for failure to pay court debts. We find, in both states, that likely on the order of 1,500 people per million residents were jailed annually for failure to pay between 2005 and 2018. In absolute terms, this translates to approximately 8,000 failure to pay jailings per year in Wisconsin and 38,000 failure to pay jailings per year in Texas. Moreover, we find that while half of court debtors are jailed for a day or less, many court debtors are jailed for days, resulting in the average court debtor spending 2.1 days in jail in Texas and 6.2 days in jail in Wisconsin. Lastly, these jail booking data indicate the existence of racial disparities in debt imprisonment, but these disparities are difficult to distinguish from disparities in jailings more generally.

In addition, this data collection effort sheds light on a number of factors that lead to debt imprisonment. Using a second dataset of Oklahoma court records, including failure to pay warrants, from between 2008 and 2018, we study the amount of fines and fees owed by court debtors. We find that while the exact amount owed varies by case type, Oklahoma court debtors whose original charge is traffic-related were assessed an average of around $250 dollars in fines, fees, and other costs; for debtors for whom judges issued FTP warrants, that number rises to around $500. We also found in all three states that traffic offenses are an important driver of these legal financial obligations. In Oklahoma, 37% of cases in which judges issued a failure to pay warrant were traffic cases; in Texas and Wisconsin, for individuals booked only for FTP, 64% of associated charges were traffic-related.

## A case study

Ms. Smith is a Black woman from Austin, Texas who was jailed for unpaid traffic citations in 2017. (Ms. Smith's name has been altered to protect her identity; research on this case study was reviewed and approved by the Stanford University IRB Administrative Panel on Human Subjects in Nonmedical Research, and written consent was obtained from research subjects.) A timeline of the events leading to her arrest is displayed in Fig 1. Ms. Smith's only criminal history is a misdemeanor theft charge stemming from a bounced $21 check for groceries. Like most residents of Austin [9], Ms. Smith's primary mode of transportation is driving. Around 2008, she encountered financial difficulties and intermittently went without auto insurance.

On May 19, 2008, Ms. Smith was pulled over by Austin police and cited for rolling through a stop sign. She was also cited for Failure to Maintain Financial Responsibility (FMFR) for driving without liability insurance (Tex. T. Code § 601.191). Under the City of Austin's current fine schedule, the base fine for FMFR is between $180 and $350. After including additional court costs, the total payment in a typical case is $357 for FMFR, and $233 for running a stop sign [10]. Under Texas's (now repealed) Driver Responsibility Program, an additional $750 in surcharges were added to the initial fine across the subsequent three years (Tex. T. Code §

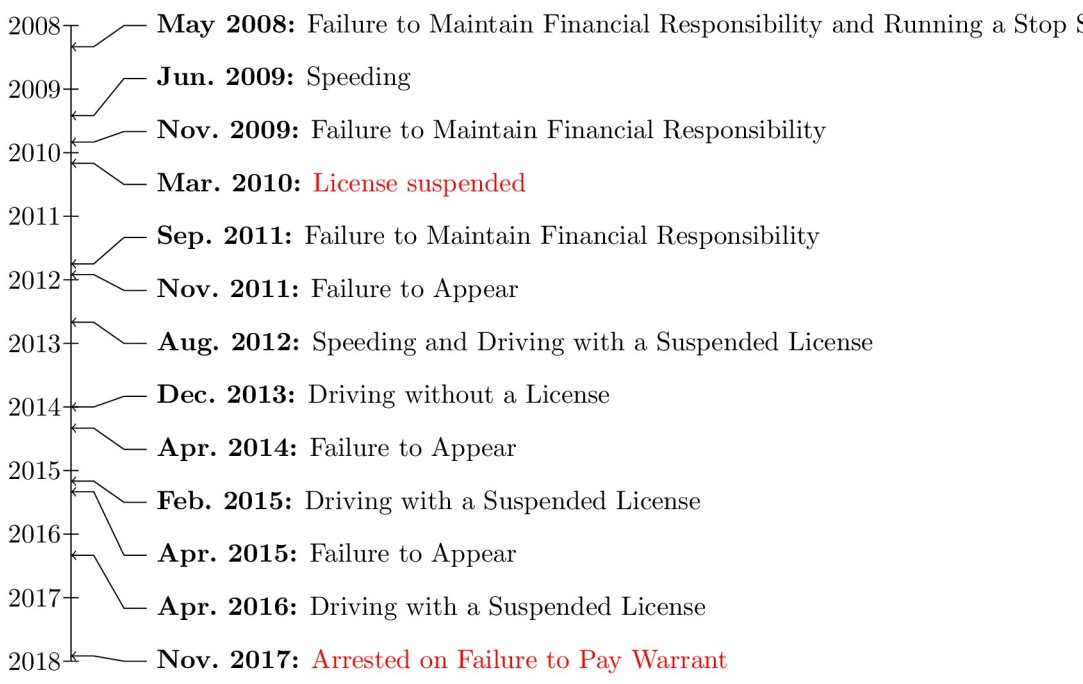

**Fig 1. Timeline of events leading to Ms. Smith's arrest for failure to pay.** Based on publicly available court records. Note that two additional citations for driving with a suspended license and failure to appear that occurred during this period on unknown dates are not shown. Neither are an additional four citations for which neither the dates of occurrence nor the underlying violations are known, but which occurred between 2011 and 2017.

708.103; repealed 2019). Individuals fined under the Driver Responsibility Program were notified of their surcharges by mail at their last known address, and if they failed to pay in full, their licenses were automatically suspended (Tex. T. Code § 708.154).

Ms. Smith did not pay the fines stemming from her 2008 FMFR citation, and her license was subsequently suspended in early 2010. Moreover, between 2009 and 2016, she accumulated additional tickets: two more for driving without liability insurance; four tickets for failing to appear at traffic court hearings; two for speeding; and several others of which we have not been able to obtain records. She also received an additional five tickets for driving with a suspended or invalid license or driving without a license. Each ticket—with a base fine of a few hundred dollars but also subject to comparable surcharges under the Driver Responsibility Program—added substantially to her court debt. By 2017, Ms. Smith owed thousands of dollars, although the exact amount is uncertain. Much of this court debt, through her suspended license, was directly traceable to her initial 2008 FMFR citation.

On November 17, 2017, Ms. Smith was pulled over driving home from work. She informed the officer that her license was suspended. After running her plates, the officer discovered that a failure to pay warrant had been issued in her name. The officer arrested Ms. Smith, took her to the Travis County Jail where she was booked for nonpayment, and impounded her car. She was pulled over around 11:00 p.m. and booked after 1:00 a.m., and was brought before a judge, put on a new payment plan, and released the following morning.

In short, Ms. Smith was jailed not because she committed a crime but rather because she failed to pay. The particular unpaid fine for which the warrant was issued came from a non-criminal traffic offense. In consequence of such offenses, Ms. Smith was forced not only to spend the night in jail but also to bear what she described as an overwhelming financial burden

for almost a decade. Her story is not unique; rather, as we show in the results section, it exemplifies some features of the larger phenomenon of debt imprisonment.

## The criminalization of poverty

Ms. Smith's jailing represents an extreme example of what some researchers have termed the "criminalization of poverty." In recent years, researchers have documented many ways in which the American criminal justice system serves to punish people for aspects of poverty, such as homelesssness [11, 12] or involvement in the welfare system [13, 14]; and through mechanisms that are disproportionately severe for poor people, such as cash bail [15] and fines and fees [16, 17]. Within the criminal justice system, researchers have also examined the roles of a variety of actors and institutions, including police [18], courts [19], and jails and prisons [20].

An important driver of this process is the growth of "legal financial obligations" or "court debts." While the criminal justice system incarcerates Americans at lower rates today than at the peak of mass incarceration roughly a decade ago [21], reliance on alternative forms of supervision and punishment, such as parole and fines and fees, has increased [22, 23]. As in the case of Ms. Smith, whose initial traffic offense occurred nearly a decade before she was ultimately jailed for failure to pay, court debts often lead to protracted entanglements with the criminal justice system, especially for the individuals least able to pay them [16, 19].

Scholars have looked extensively at why the criminal justice system imposes fines and fees so broadly. Previous research has tried to understand how a variety of factors, such as political ideology or racial threat in communities with growing minority populations, are associated with increased imposition of court debts [24, 25]. More narrowly, researchers have also examined two important potential rationales for the expansive use of fines and fees: first, that fines and fees are a deterrent to future crime; and second, that fines and fees are a kind of "reverse welfare" used to finance local and municipal governments [26–29]. The use of fines and fees as an alternative source of revenue, in particular, has been the subject of significant attention, with a number of studies finding that local governments rely more on fines and fees in times of financial distress [30, 31]. Importantly, previous work suggests that the empirical basis for both the "deterrence" and "alternative revenue source" rationales for levying heavy fines and fees are weak. A related literature has sought to understand the downstream consequences of fines and fees on court debtors, finding mixed evidence with respect to their impacts on financial well-being [32–34], but more consistent evidence that increasing such debts has little impact on recidivism [35], suggesting that the deterrence rationale may be unsupported. The imposition of court debts does, however, appear to increase the probability of future encounters with the justice system, including new court debts [35]. Likewise, the collection rate of court debts is relatively low—with estimates ranging between 11% and 24% [23, 36]—but the costs of collection relatively high, suggesting that even as a fiscal measure, levying harsh fines and fees may be counterproductive. (In some cases, counties bear the cost of jailing court debtors, while municipalities gather revenue from citations, incentivizing aggressive court debt collection policies that may be financially wasteful from a system-wide perspective; see the Discussion).

However, a great deal of uncertainty still surrounds a fundamental *descriptive* question regarding the criminalization of poverty—namely, how extensive is it? The case of Ms. Smith, and the broader phenomenon of debt imprisonment, form a distinct facet of the criminalization of poverty which is not only independently important but also comparatively straightforward to operationalize and measure. Even so, little can be said about even basic characteristics of debt imprisonment, such as how frequently it occurs, or how long court debtors typically spend in jail. Previous literature investigating debt imprisonment specifically has focused on

the *existence* of debt imprisonment, costs of collection and amounts recovered, and impacts on those imprisoned. Most of the existing research in this area is procedural, enumerating the range of fines and fees to which individuals are subject and detailing the often complex legal and policy mechanisms through which courts can jail those who fail to pay court debts [37–40], or the negative fiscal implications of aggressive court debt collection [36].

Due to the difficulty of obtaining comprehensive data on debt imprisonment, this literature has focused primarily on qualitative methods and is limited in geographic and temporal scope. Work in this vein often relies heavily on interviews and other qualitative methods to understand debt imprisonment, as in our case study above, representing a small number of individuals necessarily drawn from a single—or, at most, a handful—of jurisdictions, typically in a single state [41–44]. Perhaps the most extensive study to date, [16], uses primarily qualitative methods to look in detail at the long-lasting and disruptive consequences of monetary penalties and aggressive court debt collection on individuals in four counties in Washington. Some previous research has estimated rates of incarceration for unpaid court debt—finding that such instances comprise around 20% of jailings [8, 45]—but have examined only a handful of cities [5], often for periods of no more than a few months, making their results difficult to generalize. In line with other recent descriptive work, such as [46], a large-scale, quantitative approach using administrative data to study debt imprisonment promises not only to answer key questions about the scope and severity of debt imprisonment, but also to advance our understanding of the criminalization of poverty more generally.

## Legal background

The modern-day debt imprisonment practices investigated below occupy a legal grey area. Unlike the debtors' prisons widespread in America in the 18th and 19th century [47], "debt imprisonment" today refers to a collection of distinct legal processes sharing a common trait: after imposing a monetary penalty—also known as a "legal financial obligation" or "court debt"—the court jails an individual as a direct result of failing to pay. The nominal reason for the imprisonment can vary between states and even between different jurisdictions in the same state [48, 49]. In some states, repayment of fines and fees is a condition of parole. In others, court debtors may be presented with the option of spending time in jail to pay down legal debts (so-called "pay-or-stay" sentences), or courts may jail court debtors to compel attendance at hearings triggered by their failure to pay (FTP). Sometimes FTP is considered to be a form of contempt of court [50]. Typically, imprisonment is treated as a substitute for repayment where court debtors earn a certain amount of credit—often $50 to $100 a day (e.g., Wis. Stat. Ann. § 800.095(1)(b)(1)(a))—for each day they "sit out" their penalties in jail.

In a series of cases in the 1970s and 1980s, the U.S. Supreme Court held that imprisoning indigent court debtors who have made *bona fide* efforts to pay violates the fundamental due process and equal protection prohibitions against "punishing a person for his poverty" (*Williams v. Illinois*, 1970, [51]; *Tate v. Short*, 1971, [52]; *Bearden v. Georgia*, 1983, [4]). The Court declined, however, to prevent lower courts from jailing defendants who "willfully refused to pay" [4]. To negotiate the distinction, the *Bearden* line of cases established strong protections for court debtors, such as a right to indigency hearings and, recently, a right to representation by counsel in some cases (*Turner v. Rogers, et al.*, 2011, [53]).

Nevertheless, in many cases, policy and practice diverge. Anecdotal evidence suggests that many courts may not hold indigency hearings or may do so only perfunctorily by, for instance, assessing a defendant's ability to pay based on their clothes or whether they smoke cigarettes [16, 41, 50, 54]. A recent study of Nebraska courts found that, when assessing court debts, judges failed to inquire about defendants' ability to pay 39% of the time [55]; other studies of

individual municipalities have found rates as high as 93% [45]. This problem may be compounded by the fact that low-level traffic courts (also known as Mayor's Courts, Justice of the Peace Courts, and Municipal Courts, depending on the jurisdiction) appear to be an important driver of debt imprisonment. These local courts of limited jurisdiction help handle the petty offenses making up the bulk of the judiciary's caseload; in Wisconsin, municipal courts processed more than half of forfeiture cases—i.e., non-criminal violations—in 2016 [56, 57]. In many states, traffic court judges are often not required to have—and frequently actually lack—formal legal credentials [39, 58]. In some cases, courts with no authority to impose prison sentences can nevertheless issue arrest warrants for their debtors. Moreover, many traffic courts operate only during normal working hours and may only be accessible by car. Especially in light of the fact that traffic violations—which can result in the suspension or revocation of driver's licenses—are an important source of court debt, these factors may limit the ability of court debtors to attend indigency hearings even under ideal circumstances.

## Data and methods

### Datasets

Our results are drawn from two datasets integrated from public data sources. Our primary dataset consists of jail rosters—also variously known as "inmate booking reports" and "jail logs"—that we collected from 64 counties in Texas and 27 counties in Wisconsin, largely covering the period 2008–2018. (Some counties provided data beginning in 2005.) These jail rosters were obtained through public records requests filed with county sheriffs' offices. Our secondary dataset consists of failure to pay warrants posted online by the state of Oklahoma as part of public court records (Oklahoma State Courts Network: https://www.oscn.net/v4/). We subsequently cleaned, standardized, and anonymized the resulting data. (The data and quantitative analysis detailed here and below was based on publicly available information and exempt from IRB review. Stanford University IRB did not oversee anonymization of these data.) All clean data and the data processing code are publicly available online at https://policylab. stanford.edu/debtors-prisons. Although not included in our analyses here, we also obtained jail rosters from isolated counties in Missouri, Kentucky, and Louisiana; and warrant data for the entire state of Delaware. (Data from these states were too limited to draw meaningful conclusions about the prevalence of debt imprisonment at the state level).

In aggregate, our primary dataset is comprised of records of more than 4 million individual jail bookings. To collect these records, we submitted public records requests to county sheriffs or equivalent local law enforcement agents in all 254 counties in Texas, and all 72 counties in Wisconsin. Ultimately, 91 counties in Texas and 34 counties in Wisconsin provided records in a variety of raw formats. The contents of these records vary considerably by jurisdiction. To unify the records, we developed a record standardization pipeline, consisting of the following steps.

1. **Rectangularization**: Using a combination of open-source and bespoke tools, raw data in a variety of formats (PDF, TXT, Excel, etc.) were converted to CSVs.

2. **Classification**: The columns of the resulting CSVs were classified according to a common schema using word-frequency–based heuristics, and then verified by manual inspection.

3. **Parsing**: The contents of each column were parsed so that disparate raw values (`F`, `f`, `Female`, `WF`, etc.) would be represented according to a common system (e.g., `female` for `F`, or `white` and `female` for `WF`). The `libpostal` and `humaniformat` open source libraries were used to normalize addresses and names, respectively.

4. **Deduplication**: Disparate rows corresponding to the same booking were matched on all available common data (e.g., name, address, and booking time) and assigned a unique booking identifier.

5. **Standardization**: Entries resulting from improper parsing or clerical errors (e.g., ages greater than 100, ZIP codes not corresponding to any actual ZIP code, etc.) were dropped.

6. **Imputation**: Values which were not included in the raw data but could be calculated (e.g., age from booking date and date of birth, whether an individual had been jailed for failure to pay, etc.) were imputed.

In particular, from each booking, we attempted to extract and normalize—as available—the booked individual's race, ethnicity, age, gender, and home address or ZIP code of home address; and the date and time of the individual's booking and release. We also extracted all charges at time of booking and associated charge severities, statute citations, charging court, warrant type, bail or bond amounts, fine amounts, the reason for the individual's release, and any additional notes recorded. In cases where ethnicity was not available, we imputed it during the imputation step using surnames and any available geographic information [59]. As indicated in S1 Table, the availability of each of these aspects of jail bookings varies markedly between counties; unless otherwise indicated, each analysis below uses the full set of counties for which data sufficiently rich to carry the analysis out are available.

Our secondary data source consists of records from approximately 2.7 million court cases in Oklahoma covering all 77 of its counties. The 2.7 million traffic and criminal cases record the defendant's name and other personal information, including their address; information about the case, including filing date, presiding judge, counts, and attorneys involved in the case; and the court docket, which, importantly, records fines and fees assessed against the plaintiff, and any failure to pay warrants issued, although not whether the warrants were executed.

## Definition of "failure to pay"

Determining when an individual has been jailed solely because of failure to pay and not for some other reason raises an important methodological issue. Many people jailed on failure to pay warrants are also jailed, at the same time, for more serious offenses, including violent criminal offenses. It is possible some individuals booked for failure to pay might have been jailed even if their failure to pay warrants had been withdrawn. To conservatively estimate the number of jailings of court debtors, we consider an individual to have been jailed for failure to pay *only* if (1) charges related to failure to pay are the only ones present in the booking records provided to us—including charges for minor offenses, such as ordinance violations—and (2) taken as a whole, the booking records provided by the county where they were jailed appear to offer a reasonably complete record of all booking charges. (Many counties only provided records for individuals booked on failure to pay warrants and *only* those charges relating to failure to pay).

We identify charges related to failure to pay slightly differently, depending on the state in which the booking occurred. A number of identification criteria are shared between the states:

- **"Failure to Pay" or "FTP"**: Bookings that explicitly mention failure to pay can confidently be asserted to be related to failure to pay court-ordered fines, fees, or other legal financial obligations. The exception is bookings for failure to pay child support; therefore bookings referencing child support and related abbreviations are excluded.

- **"Laid out fines," "Pay or stay," or "X days to pay"**: These phrases are closely related to failure to pay jailings. "Laying out one's fines" refers to the practice of having court debtors

spend a fixed period of time in jail in lieu of repaying fines, fees, or other LFOs; "pay or stay" similarly refers to cases in which a court debtor is given the choice between going to jail or paying their outstanding court debt. Likewise, the fact that an individual was released with "*X* days to pay" for some number *X* indicates that they were imprisoned for failure to pay, but were then released before completely laying out their fines, with the understanding that they would pay off the remainder during some fixed period.

- **"Paid Fines" or "Fines Paid"**: Similar to "FTP," when an individual's booking indicates that they were released—or could be released—for paying their fines, it strongly indicates that the condition of their imprisonment was *non*-payment.

  In Texas we add the following criterion:

- **"Capias Pro Fine" or "CPF"**: A *capias pro fine* is a type of warrant issued in the state of Texas exclusively "after judgment and sentence for unpaid fines and costs" (Tex. C. Crim. Proc. § 43.015).

  In Wisconsin, we add the following analogous criteria:

- **"Municipal" or "Ordinance"**: In Wisconsin, municipal judges are empowered to imprison individuals appearing before them *only* in cases of failure to pay (Wis. Stat. Ann. § 800.09). Moreover, the local ordinances over which municipal courts in Wisconsin have jurisdiction cannot, by law, have any period of incarceration as punishment (Wis. Stat. Ann. §§ 66.0109, 66.0113); consequently, references to ordinance violations in jail bookings can only reflect failure to pay fines and fees associated with the underlying ordinance.

- **"Commitment"**: While a "commitment" can refer to an involuntary commitment to a jail or mental health facility for treatment, the term is commonly used in connection with commitment to jail for failure to pay (e.g., Form GF-148 [60]).

  A charge is marked as "FTP" if language meeting these criteria are found in the charge, warrant, or other booking notes associated with the charge.

## Results

### Scope and consequences of debt imprisonment

*Per capita* **booking rates.** To estimate the prevalence of debt imprisonment, out of the 64 counties in Texas for which we have clean data, we restrict to the 58 which provided sufficiently detailed jail rosters. The combined population of these counties in 2018 was 8.8 million people, or 31.6% of Texas's total population. In Wisconsin, out of the 28 counties for which we have clean data, we restrict to the 20 counties providing full jail rosters. These counties encompass 1.1 million residents, or roughly 18.2% of Wisconsin's total population in 2018. See S1 Table for complete details.

The jail booking records we received varied not only in contents but also in the period of time covered. In some instances, counties provided data covering the period 2005–2018, while most counties provided data only beginning in 2008. Not all counties provided data covering the decade beginning in 2008, and, in some cases, only provided data over intermittent periods. For that reason, our *per capita* estimates of jailings for failure to pay are calculated on a month-by-month basis, using the total number of bookings across all counties with data available in that month and the total population of those counties in the given year. (ACS five-year population estimates are only available beginning in 2009 [61–70]. ACS population estimates for 2009 were used for the years 2005–2008).

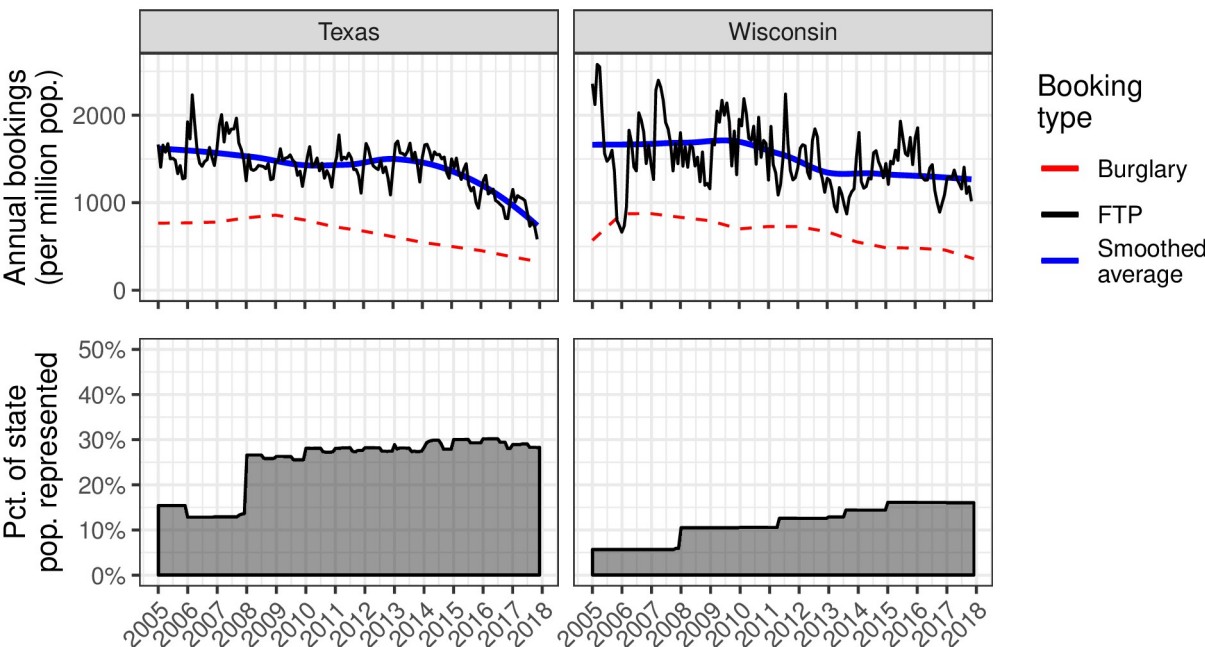

**Fig 2.** *Per capita* **booking rates for failure to pay charges alone in Texas and Wisconsin.** The top panels show the annualized *per capita* booking rates for individuals booked on *only* failure to pay charges and the bottom panels show the total population of the counties used in a given month to estimate the *per capita* rate as a percentage of the state population. The dashed red reference line in the top panels indicates the *per capita* rate of arrests for burglary in the corresponding year [71–84]. The blue line in the top panels indicates a LOESS smoothed average of the annualized FTP booking rate. The drop in Texas beginning around 2015 likely reflects policy changes that resulted in less aggressive court debt enforcement and increased requirements for indigency hearings. Counties provided data over different—sometimes non-contiguous—ranges of time. Annualized *per capita* rates for FTP bookings are calculated for each month in the period using those counties that provided data for the given month. The large jumps in the bottom panels at the beginning of 2008 are a result of the fact that many counties provided jail booking data beginning in that year.

The results are shown in Fig 2. These plots indicate that, between 2005 and 2018, the counties represented in both Texas and Wisconsin had annualized *per capita* FTP booking rates of around 1,500 bookings per million people, especially between 2005 and 2015. Averaging over the period represented, and accounting for population change, if these trends hold more widely in the two states, then we would expect roughly 8,000 bookings per year for failure to pay alone in Wisconsin and roughly 38,000 bookings per year in Texas during the period covered. In Texas, individuals jailed for failure to pay alone have, on average, 1.5 *capias pro fine* warrants when they are jailed. Between 2012 and the end of 2018, Texas municipal and justice of the peace courts issued 4.9 million *capias pro fine* warrants. Consequently, on this basis, we would estimate that roughly 8% of individuals with an active *capias pro fine* warrant were ultimately jailed on that basis alone in a given year.

We briefly note that, despite some work hypothesizing an increase in debt imprisonment following the 2008 financial crisis (e.g., [27, 48]), Fig 2 does not indicate an appreciable increase in the use of jail as a sanction in the years following the crash. Indeed, the *per capita* jailing rate for failure to pay in Wisconsin has remained relatively constant throughout the period 2005–2018. In Texas, in contrast, the *per capita* failure to pay booking rate appears to have fallen by approximately half, starting from a similar annual rate of approximately 1,500 bookings per million residents between 2005 and 2015, and falling to around 750 bookings per million residents by 2018. While this trend coincides with a larger decrease in illegal activity during the same period, the proportion of bookings for failure to pay alone among all

bookings—around 3% in Texas, as well as in Wisconsin—also decreased, suggesting a role for other factors; see S1 Fig. In particular, the decrease roughly coincides with a series of lawsuits filed against Texas government entities—including the cities of Santa Fe [85], El Paso [86], and Austin [87]—for unconstitutional debt enforcement practices; and the eventual passage SB 1913 in 2017, which made it more difficult for Texas judges to imprison court debtors for failing to pay (Tex. Crim. C. § 42.15).

We emphasize that interpreting these descriptive results—both the *per capita* rates as well as the rate at which court debtors are jailed—more generally warrants some caution. The counties included in this analysis in Texas and Wisconsin may not represent a random sample of counties in the state. Indeed, they are, for instance, less urban than the states as a whole; see S3 and S4 Figs. Moreover, there may be omissions in our data, such as individuals held for failure to pay in so-called "city jails" or "municipal lockups"—data were more difficult to obtain from these smaller facilities operated directly by municipal governments, which are not under the jurisdiction of the county sheriff or Texas Commission on Jail Standards. However, the large proportion of the population represented in our sample gives a basis for confidence that our estimate is reasonably accurate, certainly to within an order of magnitude. Additionally, the degree of agreement between this analysis and an alternative methodology using data from counties accounting for a much larger proportion of each states population provides additional evidence that our estimate is reasonably accurate; see S1 Appendix.

**Length of stay.**   To measure the length of court debtors' jail stays, we restrict to those counties providing both booking and release dates. In Texas, 57 counties, accounting for 19.4% of the state's total population, meet these criteria. In Wisconsin, 18 counties provided data including both booking and release dates, accounting for 11.5% of the state's population. Given the large number of records represented in our data, some proportion of individuals will be marked as having been arrested for failure to pay due to clerical errors. Some of the longest lengths of stay are likely a result of such clerical errors, although others may indeed simply represent exceptionally long failure to pay jailings. In the interest of conservative estimation, we exclude any bookings in which the individual in question was recorded as having spent more than 100 days in jail. This results in the loss of 0.4% of bookings in Texas and around 2.1% of bookings in Wisconsin.

The distribution of length of stay for individuals booked for failure to pay alone is given in Fig 3. In both states, most individuals are released soon after they are booked—in Texas and Wisconsin, the median length of stay is 1 day, likely reflecting the fact that in most instances, individuals arrested on an FTP warrant are brought before a judge and released the next day, as is consistent, e.g., with Texas state law (Tex. Crim. C. § 425.045). For individuals who spent less than one day in jail, it is possible that a significant portion of that time was spent in court, rather than in jail, depending on what point in the process that individual's book out time reflects. However, individuals booked for more than one day (i.e., above the median) spent at least one night in jail. We note that the distribution of stay lengths differ slightly between Texas and Wisconsin. A larger number of individuals are jailed for longer periods of time in Wisconsin than in Texas. This is reflected in the difference between the mean jail stays: 2.1 days in Texas, compared to 6.2 days in Wisconsin. The tail of longer jail stays may be driven by so-called "pay-or-stay" sentences: a higher percentage of long bookings for which the nature of the booking can be confidently determined are "pay or stay" sentences—see S2 Fig.

## Factors contributing to debt imprisonment

**Fine amounts.**   While data on fines and fees are relatively limited in booking data from Texas and Wisconsin—most jurisdictions record underlying fines only sporadically and do

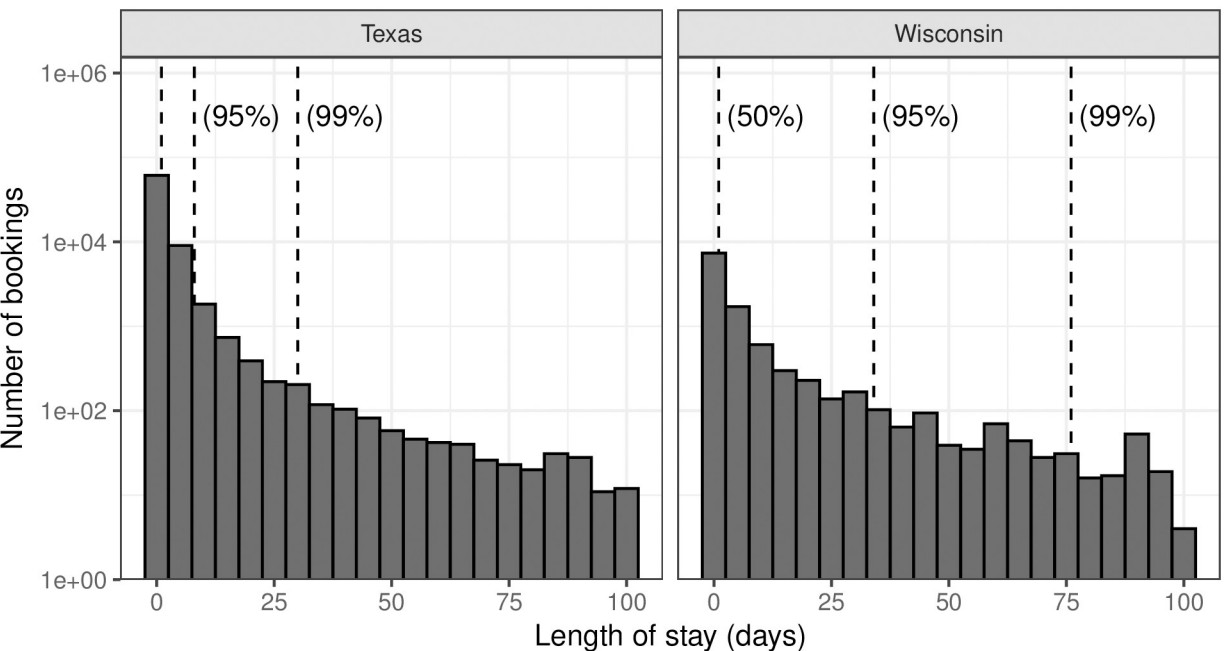

**Fig 3. Distribution of length of stay of individuals booked for failure to pay alone in Texas and Wisconsin.** These histograms show the distribution of how long individuals booked for failure to pay alone spent in jail in Texas and Wisconsin. Here the *y*-axis is on the logarithmic scale. Dashed lines indicate the median, 95th, and 99th percentiles. The distributions between the two states differ, with the distribution in Wisconsin displaying a heavier tail.

not distinguish paying fines from posting bail—court records of both FTP and non-FTP cases gathered in Oklahoma are more complete, and include itemized fines, fees, and other costs imposed on defendants.

These data, shown in Fig 4, show that fines in Oklahoma are relatively costly. This is true even after controlling for case type. In traffic cases, total costs averaged over $200 in all cases, and almost $500 in cases where a failure to pay warrant was issued. Fines imposed for criminal misdemeanors average around $1,000 overall, and nearly $1,600 for cases in which a failure to pay warrant was issued. In the case of criminal felonies, the average total cost is roughly $2,600 overall, and $3,700 in cases where a failure to pay warrant was issued. The cost of living for a single adult with no children in Oklahoma has been estimated to be roughly $22,000 per year [88], while 15.6% of the state lives below the poverty line and annual *per capita* income is estimated at only $27,000 [89, 90]. Unpaid court debt, even in traffic cases, likely represents a substantial financial shock to typical court debtors. Moreover, our data indicate that repaying court debt can be a protracted process. The median length of time between a case being filed and a failure to pay warrant being issued is around one year. For traffic cases, the median time between filing and issuing a warrant is around five months. However, this distribution has a comparatively heavy tail, and in a substantial number of cases, warrants are only issued many years after the fact: the *average* length of time between case filing and warrant issue is 1.6 years in all cases, and 0.9 years in traffic cases.

We offer two caveats regarding these results. First, the distribution of court debts in Oklahoma may differ in important ways from its distribution in Texas and Wisconsin due to policy differences between states. Previous investigations of smaller scope have found, e.g., that the median jailed court debtor in some Texas municipalities owes between $500 and $1,300, depending on location [45]. Secondly, individuals jailed for failure to pay likely differ from the

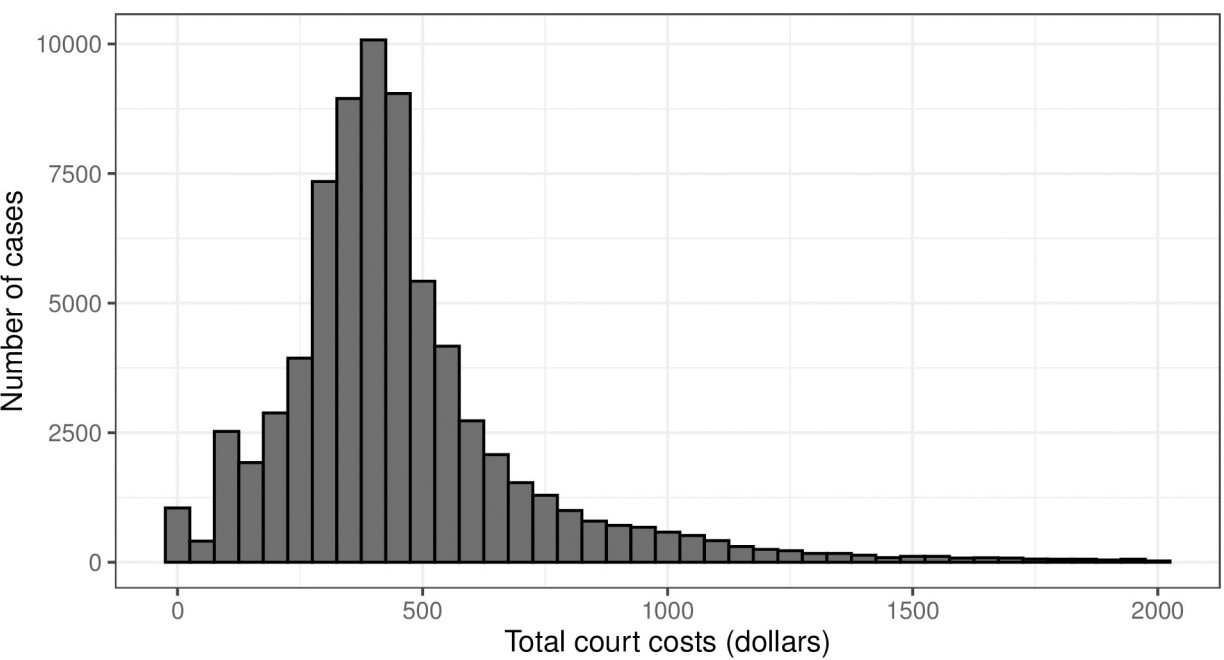

**Fig 4. Distribution of court costs in Oklahoma traffic cases in which a failure to pay warrant was issued.**

larger population of court debtors in systematic ways and, while existing literature on debt imprisonment has documented large numbers of individuals jailed for relatively small amounts of court debt (e.g., [5, 41]), it seems likely that the typical individual *jailed* for unpaid court debt owes more than the typical court debtor.

**Underlying offenses.** In Oklahoma, to a substantial degree, debt imprisonment occurs when the underlying offense is a traffic violation. While warrants are issued less frequently in traffic cases than in criminal felony or misdemeanor cases, the comparative volume of traffic cases is so high that in 37% of the cases in which a failure to pay warrant was issued, the underlying violation was a traffic violation.

This finding is broadly consistent with the charges for which court debtors are jailed in Texas and Wisconsin. Of all the charges recorded, 22% contain too little information to ascertain the nature of the charge underlying the fines that the booked individual failed to pay. (This percentage was calculated by manual classification of a random sample of 1,000 rows; see `handcount.csv` in the replication materials.) Of all the remaining charges recorded for individuals jailed for failure to pay, 64% are traffic-related. A further approximately 20% are for similarly petty offenses: around 7% relate to public intoxication, 5% to possession of marijuana or other petty drug offenses, 4% to theft, 1.7% to disorderly conduct and related offenses, and 1.7% to truancy or failure to attend school.

**Racial disparities.** Measuring racial disparities raises several conceptual issues. First, since debt imprisonment is an issue that primarily affects low-income individuals, to measure racial disparities, we benchmark jailing rates against the demographic makeup of the population living below the poverty line, rather than the total population. Second, debt imprisonment shares a number of sources of disparities in common with jail bookings more generally, including generally higher police presence in majority minority neighborhoods, discrimination in arrest and sentencing, and any differences in the underlying distribution of ordinance or other violations. To disentangle disparities unique to debt imprisonment from larger

criminal justice disparities, we compare two quantities: (1) the extent to which Black or Hispanic individuals make up the population *booked for failure to pay alone* compared to their share of the population of the county living below the poverty line; and (2) the extent to which Black or Hispanic individuals make up the population *booked for any reason* compared to their share of the population of the county living below the poverty line.

After restricting to counties and timespans with sufficiently detailed race information—that is, 57 counties in Texas, accounting for approximately 8.8 million residents and 18 counties in Wisconsin accounting for around 900,000 residents—we find that Black individuals are over-represented among all bookings (24%) and FTP bookings (29%) relative to their share of the population living below the poverty line (14%). Roughly the same pattern holds in Wisconsin, where Black individuals make up 9% of bookings for any reason and 14% of bookings for FTP, but only 2% of the population living below the poverty line in the included counties. Hispanic individuals are represented approximately in proportion to their share of the population living below the poverty line or even underrepresented: 42% of bookings for any reason and 37% of failure to pay bookings are of Hispanic individuals in Texas, compared to 54% of the population living below the poverty line in the included counties; and 6% of all bookings and 4% of FTP bookings, compared to 6% of the population in Wisconsin.

In short, our results in both Texas and Wisconsin indicate that, relative to their share of the population, Black individuals are overrepresented in FTP bookings in both states, while Hispanic individuals may in fact be underrepresented among imprisoned court debtors. However, the demographic makeup of the population of individuals booked for failure to pay alone and the population of individuals booked for any reason do not differ greatly. This suggests that these disparities are likely endemic to the criminal justice system generally, rather than specific to debt imprisonment. Finally, we note that a number of individual counties, such as Sheboygan County, Wisconsin, evidence large disparities in failure to pay jailings independent of the general trends just noted.

We note two interpretive difficulties. First, the race data present in these jail booking records reflect officer-reported rather than self-reported race, which may be inaccurate. Texas State Patrol, for instance, has erroneously reported many Hispanic drivers as being white in traffic stop data [91]. Second, since the jail population is not guaranteed to be drawn from the county at large, the benchmark demographic—the percentage of individuals living below the poverty line who are Black or Hispanic—may be inappropriate. For instance, we observe that in 39% of cases where ZIP codes are available, court debtors did not live in a ZIP code in the county where they were jailed, and that that proportion varies widely by jurisdiction; see S5 Fig. However, when we replicate our analysis, restricting only to individuals whose ZIP code matches the ZIP code of the county where they were jailed, we find results that are broadly consistent with the results above; see S2 Appendix.

## Discussion

Analyzing more than 4 million jail booking records, 2.7 million court cases, and hundreds of thousands of warrants, we have worked to quantify the scope and character of debt imprisonment. Our analysis indicates that tens of thousands of people per year were jailed for failure to pay court debts over the last decade in Texas and Wisconsin. It suggests that in a majority of cases, court debtors are spending a day or more in jail, and in some cases several days or even weeks, in addition to owing hundreds or even thousands of dollars in court debt, in many cases for relatively minor offenses, such as driving without liability insurance. Indeed, our results indicate that such low-level traffic violations are an important driver of debt imprisonment.

Together, these findings paint a picture of a system that creates heavy burdens, both financial and otherwise. Our results raise important questions about the relative harms and benefits of debt imprisonment as public policy. Courts do not recoup fines and fees from jailed court debtors, but county jails must pay to house them. A previous study of Milwaukee, Wisconsin suggests that the city of Milwaukee paid twice the amount owed in fines and fees to incarcerate individuals for failure to pay—amounts that were not recovered by jailing them [36]. Previous research has found little evidence of deterrence effects that could counterbalance this system's financial cost and the burdens it places on jailed court debtors [32, 35].

In light of this, we conclude by offering several recommendations for state and local policymakers. The example of states like Colorado, which outlawed the practice of issuing failure to pay warrants in 2014 (Col. HB 14–1061, 2014 Reg. Session), suggests that eliminating the practice of debt imprisonment entirely is a reasonable and achievable goal. Even in the absence of a complete ban on debt imprisonment, our analysis suggests steps policymakers can take to help alleviate the social and financial costs of debt imprisonment.

The first is to encourage judges to make better use of alternative sanctions instead of aggressively pursuing unpaid fines and fees. Community service remains a vastly underutilized tool in traffic courts' judicial toolkit; in Texas in 2019, LFOs in municipal court cases were satisfied at least in part by community service in only 16% of cases, and were waived in only 8% of cases [92].

Secondly, state and local lawmakers can reduce the number of failure to pay cases by simply reducing the fines and fees associated with the petty offenses like speeding that overwhelmingly account for instances of debt imprisonment. Even before accounting for surcharges like those added by the Texas Driver Responsibility Program, for people living at or below the poverty line, an unexpected $200 or $400 ticket can be a shock with destabilizing financial consequences [93]. In addition, lawmakers should be careful to eliminate sources of "fine cascades": imposing fines for charges like driving without liability insurance or driving with a suspended license that are themselves possible consequences of financial distress have the potential to create a snowball effect, rapidly multiplying the amount of unpaid fines and fees court debtors owe. In recent years, interest in "progressive" or "segmented" fines, in which individuals are fined in proportion to their wealth or income, rather than at flat rates, can normalize the deterrent effect of fines and fees across the income scale, and allow higher fines on richer individuals to subsidize lower fines on poorer individuals [94, 95]. Moreover, policymakers can make better use of non-punitive methods of promoting traffic safety, such as traffic calming measures like speed bumps and roundabouts [96]. Preventative measures such as license suspension can be targeted at improving traffic safety specifically, rather than as a means of compelling payment of court debts.

Thirdly, the financial incentives of traffic courts should be realigned. Previous research has suggested that municipal and other low-level courts may feel compelled to aggressively pursue unpaid court debts because they represent an important revenue stream for their operation [27, 41, 45]. State and local policymakers would do well to incentivize the administration of justice and disincentivize maintaining revenue neutrality. There are at least two promising avenues for accomplishing this change. First, courts could replace fine revenue with a different stream of revenue, such as pooled state funds. Second, courts could bear part of the substantial financial burden imposed on taxpayers by diverting criminal justice resources to tracking down and jailing individuals who have failed to pay court debts. Such a change is supported by evidence from other parts of the criminal justice system that aligning the incentives of prisons and courts reduces incarceration [97].

Lastly, both to promote better understanding of the extent of debt imprisonment and to help hold municipal courts and other local actors accountable, states should track the issue of

failure to pay warrants and failure to pay jailings centrally. While entities like the Texas Office of Court Administration have already taken steps to that end, the extreme difficulties we encountered both in finding jurisdictions that tracked failure to pay jailings and in convincing those jurisdictions to share that information, even with the benefit of expansive open records laws, speaks to the opaque nature of the debt imprisonment process. We believe that additional transparency will expose systemic abuses and encourage courts to imprison court debtors less frequently. In the absence of such centralized recordkeeping, we hope that this novel data collection effort will help to fill the holes in our knowledge of debt imprisonment and to reduce the social and financial costs of this practice.

## Supporting information

**S1 Appendix. Robustness checks: *per capita* rates.**
(PDF)

**S2 Appendix. Robustness checks: Racial disparities.**
(PDF)

**S1 Fig. Percentage of bookings which are for FTP alone in Texas and Wisconsin.** The solid black line indicates the percentage of bookings in the given month across all counties with applicable data in that month. (The counties included in this proportion are the same as in Fig 2, which shows *per capita* FTP booking rates.) The blue line is a LOESS smoothed average of the percentage of bookings for failure to pay alone.
(TIF)

**S2 Fig. Percentage of pay-or-stay bookings by length of stay.** The solid black line represents the percentage of bookings which, based on available information, represents a "pay-or-stay" booking among all bookings for which the length of stay was at least the indicated number of days. Pointwise 95% confidence intervals are shown in gray. We note that the indicated percentages may represent underestimates since a large proportion of bookings comprising our data set do not contain enough detail to confidently determine whether a failure to pay booking was a "pay-or-stay" booking.
(TIF)

**S3 Fig. Urbanicity of counties represented in the data.** The left-hand panels display the distribution of urbanicity of the counties in Texas and Wisconsin, according to the NCHS Urban-Rural Classification Scheme for Counties [98]. The right-hand panels show the respective distributions among counties present in the sample used to calculate *per capita* booking rates in the main text.
(TIF)

**S4 Fig. Urbanicity of counties represented in the data, weighted by population.** The left-hand panels displays the population-weighted distribution of urbanicity of the counties in Texas and Wisconsin, according to the NCHS Urban-Rural Classification Scheme for Counties [98]. The right-hand panels show the respective distributions among counties present in the sample used to calculate *per capita* booking rates in the main text, also weighted by population.
(TIF)

**S5 Fig. Percentage of individuals residing in the county in which they were jailed for failure to pay.** The counties included are those in which ZIP codes were available for the majority of jail bookings.
(TIF)

**S1 Table. Availability of fields in jail logs collected from county jails in Texas and Wisconsin.** A '•' indicates that the field is present in at least 80% of rows in the indicated county. (PDF)

## Acknowledgments

We thank Emily Lemmerman, James Hamilton, and Jane Lee for their assistance with research; Joe Nudell for his help collecting data; Joshua Grossman and Keniel Yao for assistance cleaning data; and Sophie Allen, Ravi Shroff, and Alex Chohlas-Wood for helpful discussions. Reproduction materials are available at https://policylab.stanford.edu/debtors-prisons.

## Author Contributions

**Conceptualization:** Johann D. Gaebler, Phoebe Barghouty, Cheryl Phillips, Sharad Goel.

**Data curation:** Johann D. Gaebler, Sarah Vicol.

**Formal analysis:** Johann D. Gaebler, Sarah Vicol, Sharad Goel.

**Funding acquisition:** Johann D. Gaebler, Cheryl Phillips, Sharad Goel.

**Investigation:** Johann D. Gaebler, Phoebe Barghouty, Sarah Vicol, Cheryl Phillips, Sharad Goel.

**Methodology:** Johann D. Gaebler, Sarah Vicol, Sharad Goel.

**Project administration:** Johann D. Gaebler, Cheryl Phillips, Sharad Goel.

**Resources:** Johann D. Gaebler, Phoebe Barghouty.

**Software:** Johann D. Gaebler.

**Supervision:** Johann D. Gaebler, Cheryl Phillips, Sharad Goel.

**Validation:** Johann D. Gaebler.

**Visualization:** Johann D. Gaebler.

**Writing – original draft:** Johann D. Gaebler, Sharad Goel.

**Writing – review & editing:** Johann D. Gaebler, Phoebe Barghouty, Sarah Vicol, Cheryl Phillips, Sharad Goel.

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
