## [Decision Letter · Decision Letter 0]

16 Feb 2023

PONE-D-23-00711Forgotten but not gone: a multi-state analysis of modern-day debt imprisonmentPLOS ONE

Dear Dr. Gaebler,

Thank you for submitting your manuscript to PLOS ONE. After careful consideration, we feel that it has merit but does not fully meet PLOS ONE’s publication criteria as it currently stands. Therefore, we invite you to submit a revised version of the manuscript that addresses the points raised during the review process.

Your paper benefited from two expert reviews that provided outstanding comments. I strongly urge you to provide a response to each comment. For this paper to be sufficiently revised, please pay careful attention to the methodological comments, especially important questions raised about the data processing, data composition of the sample, and measures. Moreover, the reviewers raised some insightful feedback about the interpretation of the findings.

We look forward to receiving your revised manuscript.

Kind regards,

Jim P Stimpson, PhD

Academic Editor

PLOS ONE

Journal Requirements:

2. You indicated in the ethical declaration that 'Written consent was obtained from all interviewees', yet data in your study seem to be obtained through public data requests. Can you clarify the ethical statement and update as needed? Please also state if the anonymisation of the public data was overseen/approved by IRB

Reviewers' comments:

Reviewer's Responses to Questions

**Comments to the Author**

1. Is the manuscript technically sound, and do the data support the conclusions?

Reviewer #1: Yes

Reviewer #2: Partly

2. Has the statistical analysis been performed appropriately and rigorously? 

Reviewer #1: Yes

Reviewer #2: Yes

3. Have the authors made all data underlying the findings in their manuscript fully available?

Reviewer #1: Yes

Reviewer #2: Yes

4. Is the manuscript presented in an intelligible fashion and written in standard English?

Reviewer #1: Yes

Reviewer #2: Yes

5. Review Comments to the Author

Reviewer #1: This paper's goal is to describe the scale of jail admissions caused by legal financial obligations (LFOs). For their main analyses, the authors focus on two states -- Texas and Wisconsin -- for which they obtained jail admission data. They find that jail admissions for court debt is common; and, while most stays are very short (median time is less than 1 week), there is a long tail in jail stays.

This paper builds on an existing literature documenting the prevalence of court financial obligations. The main contribution, relative to the existing literature, is to describe how frequently court financial obligations can result in jail admissions. Several recent papers have explored the scale of LFOs, the motivation behind these (see for example papers by Makowsky and co-authors cited below), and their causal effects on individual and household financial stability (Mello, 2021; Liberman, Luh and Mueller-Smith, 2022). This paper links LFOs to jail admissions. The authors conducted impressive data-collection efforts, and also made their data publicly available for future researchers to use. I also want to note the academic value of descriptive work in many areas -- and in particular, regarding the criminal justice, where much is unknown about the scale of many issues. Another recent example includes Finlay et al. (2022).

Before providing more suggestions, I want to note that Figure 2 did not appear in the draft that I was asked to review; however, it seems that Appendix Figure S2 is a robustness test for Figure 2. My comments do not account for any specificities of that figure.

My first recommendation would be for the authors to provide more information about what their "jail" variable includes. In particular, I was surprised to see that the median length of stay in both stages, is 1 day. It is my understanding that in some jurisdictions, people can come to jail voluntarily to cancel outstanding warrants; and in the data for such jurisdictions, this could appear as a jail booking. Could this also happen, at least in some counties, for people voluntarily showing up to work on a financial plan with judges? This seems very important to clarify, since it may influence the interpretation of results. This information could likely be obtained by calling courts. Relatedly, I would recommend that the authors provide a better description of what is means to have a 1-day jail stint -- does this mean 24h, so at least one night in jail? If not, I assume that much of the day was spent waiting in court to meet with a judge. In this case, this paper would be closer to those in Pager et al (2022), who find no effects of court-related fee relief on jail admissions, but do find that it influences future criminal justice contacts. On the other side of the distribution, I would have liked to see more details on what drives long (or very long) jail stays -- are these mainly "pay or stay" cases, or what could be driving these very long jail stays? This also seems important to understand, especially from a policy perspective.

I also had some recommendations in terms of what analyses to present:

1. In figure 3, I would recommend that the authors plots directly the median and 75th and / or 90th percentiles of length of stay. I would also recommend presenting, in the abstract, median rather than mean length of stay figures -- especially since the high mean is driven by some very long stays (which again, the authors should explain more).

2. I appreciated the inclusion of benchmarks for the scale of jail exposure due to LFOs. A few more comparisons that would be helpful to include, if possible, are: (1) in addition to jail bookings, percent of the jail population at a given point in time incarcerated because of LFOs (which I assume will be quite small, since length of stay is small); and (2) Percent of people with LFOs who are incarcerated. It would also be informative to compute this broken down by racial groups. This second metric may be harder to compute, since it is not clear what the right timeframes to consider LFO issuances is; but for example, if the authors have the date of the LFO leading to a jail booking, perhaps using other data collected by the Stanford Open Policing Project could help obtain a good initial pool?

3. I wasn't sure how informative the results on the distribution of fines was (presented just for Oklahoma), since the sample of people with FTP warrants may be quite different from those incarcerated. For example, I am not sure that the smaller amounts (the average court debt for those with a traffic violation and an FTP warrant is around $500) would lead to a jail booking. I would recommend that the authors try to tie this better to the main results.

Lastly, I would encourage the authors to engage some more with the recent experimental and quasi-experimental literature seeking to estimate the causal effects of court debt on later outcomes. I think that Pager et al (2022) is the only paper to directly consider jail bookings as an outcome; but for example, Liberman et al (2022) find no or small effects of fines and fees on economic stability. The goal of the present paper is different, and I want to reiterate that I think there is great value in this kind of qualitative work; but tying it better to this literature would help clarify the contribution.

References:

Finlay, Keith, Michael Mueller-Smith, and Brittany Street. "Measuring intergenerational exposure to the US justice system: Evidence from longitudinal links between survey and administrative data." Working paper (2022).

Lieberman, Carl, Elizabeth Luh and Michael Mueller-Smith. “Criminal court fees, earnings, and expenditures: A multi-state RD analysis of survey and administrative data.” Working paper (2022).

Makowsky, Michael D. "Revenue-Motivated Law Enforcement: Evidence, Consequences, and Policy Solutions." (2021)

Makowsky, Michael D., and Thomas Stratmann. "Political economy at any speed: what determines traffic citations?." American Economic Review 99.1 (2009): 509-527.

Makowsky, Michael D., Thomas Stratmann, and Alex Tabarrok. "To serve and collect: the fiscal and racial determinants of law enforcement." The Journal of Legal Studies 48.1 (2019): 189-216.

Mello, Steven. "Fines and financial wellbeing." Working paper (2021).

Reviewer #2: This is a very important issue that is under-studied and requires data that is difficult to collect in a standardized fashion. I commend the authors' efforts to address debt imprisonment. However, I would like a bit more context on existing research and theory on this topic. For example, the authors state that a study was completed in Rhode Island, but provide no details on the study's findings.

I would also like to see a bit more discussion about the rationales for fines and fees: 1) deterrence and 2) to fund local government. The logic of the former is obvious. Regarding the latter, I have read significant anecdotal evidence that units of local government are largely funded on the backs of offenders...particularly in Southern regions of the country. However, is there any systematic analysis of how fines and fees are used directly or indirectly to fund local government? At the end of the document, the authors should spend more time outlining how the current findings support, reject, or expand upon these rationales. As a side note, I constantly hear the debate between the importance of deterrence versus rehabilitation concerning more serious crimes, but we never talk about rehabilitation when it comes to minor crimes and traffic violations that result in fines and fees.

I would like more details on how county data in "raw formats" was assessed and cleaned to develop the data sets. My experience is that this is an incredibly challenging task and I believe it would behoove the broader field of research in criminal justice and corrections to collaborate on how we address this issue.

The authors indicate that data from city jails might be omitted. Does this mean that rural areas were more likely to have their data included in the data sets? Our experience is that data from rural areas is much less automated than data from urban jurisdictions. Was this the case in the current data collection? If so, how was this addressed? My anecdotal knowledge is that rural jurisdictions rely more on fines and fees for their budgets, as compared to more urban jurisdictions. Do the current findings support that assertion? In short, I would like more information about the rural/urban makeup of the data sets, rural/urban differences in data collection, and urban/rural differences in findings.

The "denominator problem" is a serious problem for looking at racial/ethnic disparities in traffic stops. It is also a major issue for this analysis as those receiving traffic violations might be "passing through" rather than live in the jurisdiction of study. Are you able to analyze the data separately for cases that are or are not related to traffic violations to see if there are changes in the racial/ethnic disparities?

Finally, and related to my earlier point about deterrence and rehabilitation, for whatever reason, speeding has become a more serious problem in my state of residence. The State Patrol has seen huge increases in traffic violations for individuals traveling over 100 mph, causing serious concerns for traffic safety. Fines are our primary mechanism to attempt to deter speeders. Although I agree that fines are problematic to the extent that the lead to debt imprisonment, what is an alternative for deterring traffic violations? In my criminology courses, I have introduced Scandinavian methods of adopting progressive fines for speeding, such that rich people could be fined thousands of dollars for a simple speeding violation (which blows the minds of students). High fines on the rich can subsidize smaller fines on the poor. I would be interested in seeing more proposed solutions to the issues addressed in this paper.

6. PLOS authors have the option to publish the peer review history of their article (what does this mean?). If published, this will include your full peer review and any attached files.

Reviewer #1: No

Reviewer #2: **Yes: **Ryan Spohn

---

## [Author Response · Author response to Decision Letter 0]

9 Jul 2023

Full responses to reviewer comments are contained in the "Response to Reviewers" document.

We have clarified which aspects of this study were subject to IRB approval.

We have conformed the manuscript to the PLOS ONE formatting guidelines, including filename conventions.

---

## [Decision Letter · Decision Letter 1]

8 Aug 2023

Forgotten but not gone: a multi-state analysis of modern-day debt imprisonment

PONE-D-23-00711R1

Dear Dr. Gaebler,

We’re pleased to inform you that your manuscript has been judged scientifically suitable for publication and will be formally accepted for publication once it meets all outstanding technical requirements.

Kind regards,

Jim P Stimpson, PhD

Academic Editor

PLOS ONE

Additional Editor Comments (optional):

Reviewers' comments:

Reviewer's Responses to Questions

**Comments to the Author**

1. If the authors have adequately addressed your comments raised in a previous round of review and you feel that this manuscript is now acceptable for publication, you may indicate that here to bypass the “Comments to the Author” section, enter your conflict of interest statement in the “Confidential to Editor” section, and submit your "Accept" recommendation.

Reviewer #2: All comments have been addressed

2. Is the manuscript technically sound, and do the data support the conclusions?

Reviewer #2: Yes

3. Has the statistical analysis been performed appropriately and rigorously? 

Reviewer #2: Yes

4. Have the authors made all data underlying the findings in their manuscript fully available?

Reviewer #2: Yes

5. Is the manuscript presented in an intelligible fashion and written in standard English?

Reviewer #2: Yes

6. Review Comments to the Author

Reviewer #2: (No Response)

7. PLOS authors have the option to publish the peer review history of their article (what does this mean?). If published, this will include your full peer review and any attached files.

Reviewer #2: **Yes: **Ryan Spohn

---

## [Editor Report · Acceptance letter]

21 Aug 2023

PONE-D-23-00711R1 

Forgotten but not gone: a multi-state analysis of modern-day debt imprisonment 

Dear Dr. Gaebler:

I'm pleased to inform you that your manuscript has been deemed suitable for publication in PLOS ONE. Congratulations! Your manuscript is now with our production department. 

Kind regards, 

on behalf of

Prof Jim P Stimpson 

Academic Editor

PLOS ONE